# Loss of CDKN1A mRNA and Protein Expression Are Independent Predictors of Poor Outcome in Chromophobe Renal Cell Carcinoma Patients

**DOI:** 10.3390/cancers12020465

**Published:** 2020-02-17

**Authors:** Riuko Ohashi, Silvia Angori, Aashil A. Batavia, Niels J. Rupp, Yoichi Ajioka, Peter Schraml, Holger Moch

**Affiliations:** 1Histopathology Core Facility, Faculty of Medicine, Niigata University, Niigata 951-8510, Japan; riuko@med.niigata-u.ac.jp (R.O.); ajioka@med.niigata-u.ac.jp (Y.A.); 2Department of Pathology and Molecular Pathology, University and University Hospital Zurich, Zurich CH-8091, Switzerland; Silvia.Angori@usz.ch (S.A.); Aashil.Batavia@usz.ch (A.A.B.); niels.rupp@usz.ch (N.J.R.); holger.moch@usz.ch (H.M.); 3Division of Molecular and Diagnostic Pathology, Graduate School of Medical and Dental Sciences, Niigata University, Niigata 951-8510, Japan

**Keywords:** chromophobe renal cell carcinoma, copy number loss, CDKN1A expression, patient survival, prognosis

## Abstract

Chromophobe renal cell carcinoma (chRCC) patients have good prognosis. Only 5%–10% patients die of metastatic disease after tumorectomy, but tumor progression cannot be predicted by histopathological parameters alone. chRCC are characterized by losses of many chromosomes, whereas gene mutations are rare. In this study, we aim at identifying genes indicating chRCC progression. A bioinformatic approach was used to correlate chromosomal loss and mRNA expression from 15287 genes from The Cancer Genome Atlas (TCGA) database. All genes in TCGA chromophobe renal cancer dataset (KICH) for which a significant correlation between chromosomal loss and mRNA expression was shown, were identified and their associations with outcome was assessed. Genome-wide DNA copy-number alterations were analyzed by Affymetrix OncoScan^®^ CNV FFPE Microarrays in a second cohort of Swiss chRCC. In both cohorts, tumors with loss of chromosomes 2, 6, 10, 13, 17 and 21 had signs of tumor progression. There were 4654 genes located on these chromosomes, and 13 of these genes had reduced mRNA levels, which was associated with poor outcome in chRCC. Decreased CDKN1A expression at mRNA (*p* = 0.02) and protein levels (*p* = 0.02) were associated with short overall survival and were independent predictors of prognosis (*p* < 0.01 and <0.05 respectively). CDKN1A expression status is a prognostic biomarker independent of tumor stage. CDKN1A immunohistochemistry may be used to identify chRCC patients at greater risk of disease progression.

## 1. Introduction

Chromophobe renal cell carcinoma (chRCC) is the third most common histological subtype of RCC and accounts for approximately 5–7% of RCC [1,2,3]. Although chRCC patients have better prognoses than patients with clear cell RCC (ccRCC) or papillary RCC (pRCC) [1,2,3,4,5], about 5–7% of patients die of metastatic disease [4,6,7]. Therefore, it is of utmost importance to identify prognostic factors, which can better predict the small patient group with clinical progression after surgical resection.

The current 2016 World Health Organization (WHO)/International Society of Urological Pathology (ISUP) grading system and the older Fuhrman grading are not recommended for chRCC [1,8], although several studies have challenged to develop a histopathological grading system for chRCC [4,6,7,9,10,11,12,13]. Therefore, chRCCs are currently not graded. Interestingly, only recently, it was reported that classic chRCC harbors a larger number of chromosomal losses than in the eosinophilic subtype [14], which is often accompanied by reduced expression of “CYCLOPS” (Copy-number alterations Yielding Cancer Liabilities Owing to Partial losS)” genes [15].

Recent comprehensive genomic analyses of two chRCC cohorts demonstrated a low exonic somatic mutation rate in these tumours and identified *TP53* (20–32%) and *PTEN* (6–9%) as the most frequently mutated genes [16,17]. Casuscelli et al. [7] found increased mutation rates in *TP53* (58%) and *PTEN* (24%) as well as imbalanced chromosome duplication (≥ 3 chromosomes, 25%) in chRCC patients with metastatic disease. As the prognostic relevance of these genomic alterations was analyzed separately, the combinatorial impact of these parameters remained unclear.

In this study, we aimed to identify molecular alterations associated with survival in chRCC. We analyzed the The Cancer Genome Atlas (TCGA) Kidney Chromophobe (KICH) database [16] and a Swiss chRCC cohort for chromosomal copy number variation (CNV). Next, we focused on genes, whose mRNA expression correlated with copy number (CN) loss of chromosomes 2, 6, 10, 13, 17 and 21. Reduced CDKN1A mRNA and protein expression levels were associated with poor outcome in chRCC.

## 2. Results

### 2.1. Chromosomal Loss and Patient Outcome

The loss of one copy of chromosomes 1, 2, 6, 10, 13, 17, 21 and Y occurs in the majority of chRCC cases. Since losses of chromosomes 1 and Y have been reported in benign oncocytoma [5,16,18,19], we speculated that only loss of chromosomes 2, 6, 10, 13, 17 and 21 may be associated with outcome in chRCC patients. The frequencies of loss of these chromosomes were similar in both the TCGA-KICH and the Swiss cohort. The data are summarized in Appendix A. As recently described by our group [14], CN loss of chromosome 2, 6, 10, 13, 17, and 21 in single analysis is not associated with worse survival (Appendix A). In contrast, tumors without loss of chromosomes 2, 6, 10, 13, 17 and 21 had 100% survival in both, the TCGA and Swiss cohort (Figure 1). 

### 2.2. Identification of Genes Associated with Chromosomal Loss, Decreased Expression and Patient Survival

In search of molecular prognostic markers, we hypothesized that the expression of several genes located on chromosomes 2, 6, 10, 13, 17 and 21 is influenced by allele loss, which may affect prognosis of chRCC. The strategy to identify such genes is presented in Figure 2 and described in detail in the Materials and Methods section. The 13 candidate genes associated with chromosomal loss, decreased expression and patient survival in chRCC according to combination of UALCAN [20,21] and the Human Protein Atlas [22,23] websites are listed in Table 1 and Appendix A. Scatter plots showing the correlation between CNV and mRNA expression levels of the 13 genes according to the analyzed result acquired from the Broad Institute FIREHOSE [24] website are presented in Appendix A. mRNA expression levels of the 13 genes in normal tissue and tumors with CN loss and no loss are illustrated in Appendix A. We performed also Protein–Protein Interaction Networks Functional Enrichment Analysis using the STRING database to find interactions and pathways shared between the 13 genes/proteins. The interaction network of the 13 genes is illustrated in Appendix A. We observed strong interactions between FBXW4 (F-Box and WD Repeat Domain Containing 4), FBXL15 (F-Box and Leucine Rich Repeat Protein 15) and SOCS3 (Suppressor of cytokine signaling 3) and a weaker interaction between KLF6 (Krueppel-like factor 6) and CDKN1A. According to the Reactome Pathway Database FBXW4, FBXL15 and SOCS3 are involved in ubiquitination. Interestingly, KLF6 activates CDKN1A transcription independent from TP53 and is frequently downregulated in human tumors [25].

### 2.3. CDKN1A mRNA and Protein Expression

Among these 13 genes, we focused on *CDKN1A* whose gene product acts as a cell cycle regulator being involved in genomic stability [28] for the following reasons: (i) according to the Human Protein Atlas database for several proteins (PAOX, FBXL15, RAB37, C21orf2) antibodies suitable for immunohistochemical staining are not available or unspecific; (ii) all proteins but CDKN1A are either not or only weakly expressed in renal cell carcinoma, which significantly hampers reliable TMA expression analysis.

In the TCGA-KICH cohort, tumors with high *CDKN1A* mRNA expression separated by both the best separation cutoff (*p* = 0.02, log rank test, Figure 3A and Appendix A) and median expression (*p* = 0.026, Appendix A) had a significantly better prognosis than tumors with low *CDKN1A* mRNA expression. 

In parallel, we examined CDKN1A protein expression in 57 Swiss chRCCs by immunohistochemistry (IHC). All normal renal cells including glomeruli, renal tubules, endothelial cells, fibroblasts, inflammatory cells were CDKN1A negative (*n* = 46), with the exception of a few tubules with very weak nuclear CDKN1A staining (Figure 4A). CDKN1A-positive clear cell RCC from a previous study served as positive controls (Figure 4B) [29]. A representative image of CDKN1A-positive chRCC is shown in Figure 4C and Appendix A. An amount of 30 chRCCs (52.6%) were CDKN1A negative, 27 tumors (47.4%) were CDKN1A positive (cut off ≥ 2% tumor cells). There was a significant correlation between CDKN1A negativity and shorter overall survival (Figure 3B).

Nuclear staining was weak in 19 (70.4%) tumors and 8 (29.6%) showed moderate to strong nuclear staining. The mean (range) of the H-score (described in Materials and Methods) among CDKN1A positive tumors was 16.6 (2–110) (Appendix A). Neither staining intensity nor H-score (>20) improved overall survival rate. Nuclear staining with any intensity and a cutoff of ≥2% positive tumor cells proved to be the best criteria to differentiate between CDKN1A expression status and patient outcome.

### 2.4. CDKN1A Expression, Tumor Stage, Grade and Outcome

Analysis of the TCGA and the Swiss cohort revealed no correlation between CDKN1A expression (RNA and protein) and tumor stage. Univariate Cox regression analysis showed that both T stage (*p* = 0.004) and low *CDKN1A* mRNA expression (*p* = 0.001) were significant prognostic factors in the TCGA-KICH cohort (Table 2). In the Swiss dataset, only the absence of CDKN1A protein expression by IHC was significantly associated with poor outcome (*p* < 0.05), whereas advanced pT stage did not correlate with survival by univariate Cox regression analysis. A recently published two-tiered grading system was available for the TCGA-KICH cohort [30] and included in our calculations. Univariate Cox regression analysis demonstrated strong prognostic relevance of this grading system (*p* < 0.001) (Table 2).

Multivariate analysis using Cox proportional hazard model revealed T stage (*p* = 0.012), grade (*p* = 0.017) and low *CDKN1A* mRNA expression (*p* = 0.026) as significant independent predictors of poor outcome in the TCGA cohort. In the Swiss dataset, only loss of CDKN1A expression (*p* < 0.05) was confirmed as a significant independent predictor of poor outcome (Table 2).

## 3. Discussion

In this study, we attempted to identify molecular biomarkers with prognostic value in chRCC. For this purpose, we screened TCGA-KICH data to extract genes located on frequently deleted chromosomes whose expression is associated with patient outcome. Tumor suppressor Cyclin-dependent kinase inhibitor 1A (*CDKN1A*) was among 13 genes which fulfilled these criteria. We demonstrated that decreased CDKN1A expression at the mRNA and protein levels is an independent predictor of outcome in two independent chRCC cohorts.

The tumor suppressive role of CDKN1A, also known as p21/Waf1/Cip1, has been widely accepted. Cellular stressors, such as DNA damage or UV-light, activate tumor suppressor p53, which leads to the transient expression of CDKN1A. CDKN1A inhibits cyclin-CDK1, -CDK2, and CDK4/6, which regulates cell cycle progression of G1 and S phases and mediates senescence or apoptosis [28]. Previous studies emphasize CDKN1A’s important tumor suppressive role by showing that its depletion in cell line models leads to DNA damage and chromosomal instability [28,31] but also permits carcinogenesis from chronically damaged kidney epithelial cells [32].

*CDKN1A*, which resides in 6p21.2, is affected by the frequent loss of one chromosome 6 allele in chRCC. Analysis of TCGA-KICH data demonstrated that the loss of one *CDKN1A* allele was closely linked to lower *CDKN1A* mRNA expression levels compared to tumors that retained both *CDKN1A* alleles. Notably, the overall mRNA expression level in normal renal tissue was higher than in chRCC with *CDKN1A* loss and lower than in tumors without *CDKN1A* loss. This is consistent with the immunohistochemical CDKN1A protein expression analysis of the Swiss cohort. chRCC cells were either CDKN1A negative or strongly positive. Nuclei of glomeruli, endothelial cells, and fibroblasts were negative in the normal kidney. Only some tubular cells had weak CDKN1A expression.

Like *CDKN1A* on chromosome 6—which is absent in 80% of chRCC—the tumor suppressor genes *PTEN* and *TP53* are located on chromosomes (chromosome 10 and 17) that are also frequently lost in chRCC. Whereas *PTEN* and *TP53* are mutated in up to 9% and 32% of chRCC [16,17], respectively, *CDKN1A* gene mutations are rare [16,33]. Although immunohistochemical analysis showed no correlation between CDKN1A, TP53 and PTEN expression in chRCC (TP53 and PTEN positivity was rare in our chRCC cohort; data not shown), the loss of function of the latter two tumor suppressors may have significant impact on *CDKN1A* regulation. One important downstream target of TP53 is *CDKN1A* [34]. The downregulation of CDKN1A may thus be caused through loss of functional TP53 in those chRCC in which *TP53* is inactivated by two hits, chromosomal loss and mutation. In addition, it was shown that interaction between PTEN and TP53 stimulates TP53-mediated transcription and stabilizes TP53 [35,36,37]. In a minor fraction of chRCC loss of PTEN function may therefore exert similar negative effects on CDKN1A expression. It is tempting to speculate that a combination of loss of chromosomes 6, 10, and 17 and molecular two-hit disruption of *PTEN* and *TP53* are the main drivers for the loss of CDKN1A expression and worse patient outcomes in chRCC.

Importantly, our survival analysis revealed a clear association between reduced *CDKN1A* mRNA expression levels and CDKN1A immuno-negativity with worse outcome. Data on the prognostic relevance of CDKN1A expression are controversial in the literature and seem to be dependent on cancer type. Increased CDKN1A levels are associated with poor outcome in esophageal, ovarian, prostate cancers as well as in gliomas [38,39,40,41,42,43], while loss of CDKN1A expression is associated with decreased survival in breast, cervical, gastric, and ovarian cancers [44,45,46,47]. In some cancers, the loss of CDKN1A expression upregulates genes that repress *CDKN1A* transcription, such as *MYC* [25,48]. Ubiquitin-dependent and -independent proteosomal degradation of CDKN1A may also contribute to tumorigenesis [25,49]. CDKN1A can also exhibit oncogenic activities in some cancers, which may explain the strong correlation of its overexpression with tumor grade, rapid progression, poor prognosis, and drug resistance [25,28,43,50]. This two-faced nature of CDKN1A seems to be dependent on its cellular location. Several IHC studies imply that nuclear expression of CDKN1A indicates its tumor-suppressive role, while its presence in the cytoplasm favors an oncogenic role [25,51,52,53,54]. We have observed a significant correlation between CN loss, decreased CDKN1A expression and poor prognosis, suggesting a tumor suppressive role of CDKN1A in chRCC. This is supported by the strong CDKN1A positivity seen in tumor cell nuclei of almost half of the analyzed chRCC.

Our proposed data mining strategy demonstrated its usefulness to identify expression patterns of 13 candidate genes with prognostic impact in chRCC. However, the validation of gene expression data using additional and independent patient cohorts and different technological platforms is of utmost importance to confirm the robustness of the data. Due to the lack of suitable antibodies and only low protein expression levels in RCC, we decided to forego an immunohistochemical in situ analysis of 12 of 13 candidates. In contrast to genes and proteins that are highly differentially expressed in cancer, the validation of low abundance genes as diagnostic and prognostic tools in tumor pathology is a big challenge. Branched probe-based or enzymatic amplification RNA-ISH methods for the detection and quantification of transcripts in FFPE tissues [55] may be ideally suited to evaluate cancer biomarker candidates on the mRNA level. Given the huge amount of survival-related gene expression data in the TCGA database, systematic and comprehensive gene expression profiling of such candidate genes are necessary to better understand the complex regulatory network along tumor progression, which may lead to new therapeutic strategies to treat aggressive chRCC.

From a clinical viewpoint time to progression or tumor-specific rather than overall survival after tumorectomy are the most important parameter for chRCC [30]. Biomarkers, which predict time to progression are therefore highly desirable to identify approximately 5%–10% of chRCC at risk for progression. Additional chRCC cohorts are needed to validate whether the loss of CDKN1A expression is a reliable molecular marker to detect chRCC patients with at greater risk of disease progression.

## 4. Materials and Methods 

### 4.1. Data Acquisition and Processing Using the Cancer Genome Atlas Data Portal

Digital whole slide images of TCGA-KICH cases were reviewed using the Cancer Digital Slide Archive [56]. The corresponding clinical information of TCGA-KICH was obtained from the TCGA Data Portal [57]. Publically available Level 3 TCGA datasets comprising 66 primary chRCCs (TCGA-KICH) were downloaded from the Broad Institute TCGA Genome Data Analysis Center via FIREHOSE [24] including GISTIC copy number (CN) data, Next Generation Sequencing (NGS)-based whole genome sequencing data and RNA-sequencing data as previously described [14,15,16,58]. Two patients with missing or too short follow-up (less than 30 days) were excluded from the Cox regression analysis. TCGA CNV analysis was performed with Affimetrix SNP 6.0 with cutoff value −0.1 for copy number loss according to the Broad Institute FIREHOSE website description [24]. Gene expression values were log2-transformed to plot *CDKN1A* mRNA expression profiles of normal kidney and tumors with and without CN loss.

In the TCGA-KICH cohort, the median age at diagnosis was 49.5 years (range 17–86 years). The median follow-up of the entire cohort was 80.5 months. Nine patients (14.1%) died during follow-up. Forty-five chRCC were early stage (T1 and T2) and 19 late stage tumors (T3 and T4).

### 4.2. Strategy for Gene Candidate Selection

In a first step we used the Broad institute FIREHOSE website (“Correlate CopyNumber ys mRNAseq”) [24] to download all 15,287 available human genes of the whole genome and extracted 4654 with significant positive correlation between gene copy number and mRNA expression (Pearson’s correlation coefficient R > 0 and *p* < 0.005).

1631 of the 4654 genes were located on chromosomes 2, 6, 10, 13, 17 and 21. Since Figure 1 demonstrated chromosomal loss in 84% (79 of 94) chRCC, we hypothesized that by using a two-tiered separation based on presence or absence of chromosomal losses, the expression patterns of several genes on chromosomes 2, 6, 10, 13, 17 and 21 would fulfill the UALCAN [20,21] survival curve separation criteria: patients with high gene expression values > 3rd quartile versus patients with low gene expression (<3rd quartile). Obtaining survival curves separated by mRNA expression level in UALCAN [20,21] requires only minimal steps among three websites, UALCAN [20,21], the Human Protein Atlas [22,23] and FIREHOSE [24]. We entered all 1631 gene symbols in input fields of the UALCAN [20,21] and extracted the genes of > 3rd quartile high gene expression group with more than 80% overall survival rate. Next, we selected genes, of which the low gene expression was significantly correlated with poor prognosis (*p* < 0.05) and high mRNA expression group showed >80% overall survival rate using the Human Protein Atlas [22,23] (Appendix A). Finally, 13 candidate genes were identified (Table 1).

### 4.3. Swiss Chromophobe Renal Cell Carcinomas

A total of 57 chRCCs were retrieved from the archives of the Department of Pathology and Molecular Pathology of the University Hospital Zurich (Zurich, Switzerland). Overall survival data were obtained from the Zurich Cancer Registry. The study was approved by the Cantonal Ethics Committee of Zurich (BASEC-No_2019-01959) in accordance with the Swiss Human Research Act and with the Declaration of Helsinki. All tumors were reviewed by two pathologists (Riuko Ohashi and Holger Moch) blinded to clinico-pathological information. The tumors were histologically classified according to the WHO classification [1]. In the Swiss cohort, the median age at diagnosis was 62 years (range 18–87 years). The median follow-up was 51.0 months and 14 patients (24.6%) died during follow-up. Tumors were staged according to the TNM staging system [59]. A total of 48 chRCC were early stage (T1 and T2) and 9 late stage tumors (T3 and T4). 

### 4.4. OncoScan Assay

DNA from formalin-fixed, paraffin-embedded (FFPE) tumor tissue samples was obtained by punching 4 to 6 tissue cylinders (diameter 0.6 mm) from each sample. Punches were taken from tumor areas displaying >90% cancer cells which were marked previously on Hematoxylin and Eosin stained slides. DNA extraction from FFPE tissue was done as previously described [14,15,60]. The double-strand DNA (dsDNA) was quantified by the fluorescence-based Qubit dsDNA HS Assay Kit (Thermo Fisher Scientific, Inc., Waltham, MA, USA) according to manufacturer’s instructions. Thirty chRCCs had sufficient DNA quality for copy number analysis. Genome-wide DNA copy-number alterations were analyzed by Affymetrix OncoScan^®^ CNV FFPE Microarrays (Affymetrix, Santa Clara, CA, USA) as previously described [14,15,61]. The samples were processed by IMGM Laboratories GmbH (Martinsried, Germany). The data were analyzed by the OncoScan Console (Affymetrix) and Nexus Express for OncoScan 3 (BioDiscovery, Inc. El Segundo, CA, USA) software using the Affymetrix TuScan algorithm. The CNV cutoff value was -0.3 for copy number loss in Nexus Express for OncoScan 3 Software (BioDiscovery) default setting.

### 4.5. Immunohistochemistry

A tissue microarray (TMA) with 57 chRCC was constructed as described [29,62]. TMA sections (2.5μm) were transferred to glass slides and subjected to immunohistochemistry using Ventana Benchmark XT automated system (Roche Diagnostics, Rotkreuz, Switzerland). CDKN1A was immunostained using polyclonal anti-rabbit sc-397 (dilution 1:50; Santa Cruz Biotechnology, Inc.; Dallas, TX, USA). Immunostained slides were scanned using the NanoZoomer Digital Slide Scanner (Hamamatsu Photonics K.K., Shizuoka, Japan). Immunohistochemical evaluation was conducted by two pathologists (R.O. and H.M.) blinded to the clinical data. The criteria for protein expression analysis were as described in previous TMA studies [15,29]. A tumor was considered CDKN1A positive if ≥ 2% of the tumor cells showed unequivocal nuclear expression. A semi-quantitative approach (H-score) was also performed. The staining percentages (range 0–100%) and the intensity of nuclear expression of CDKN1A (range 0–3: 0, negative; 1, weak; 2, moderate; and 3, strong) in tumor cells were evaluated and the H-score was calculated using the formula 1 × (% of 1+ cells) + 2 × (% of 2+ cells) + 3 × (% of 3+ cells) (giving a score that ranged from 0 to 300) [63]

### 4.6. Statistical Analysis

All statistical analyses were conducted using R, 3.4.1 (R Foundation for Statistical Computing, Vienna, Austria) and EZR, Version 1.37 (Saitama Medical Center, Jichi Medical University, Saitama, Japan) [64]. The Fisher’s exact test was used to assess association between two categorical variables. Overall survival rates were determined according to the Kaplan–Meier method and analyzed for statistical differences using a log rank test. Univariate and multivariate analyses were performed by using the Cox-proportional hazard model with Firth’s penalized likelihood [65,66]. Cox regression analysis was performed using FIREHOSE mRNA expression data [24]. *p*-values < 0.05 were regarded as statistically significant.

## 5. Conclusions

In conclusion, chRCC without loss of chromosomes 2, 6, 10, 13, 17 and 21 have a favorable prognosis. CDKN1A mRNA and protein expression levels were of prognostic relevance independent from tumor stage. CDKN1A IHC is easily applicable in routine pathology and will help to stratify chRCC patients that have a significantly greater risk of disease progression. 

## Figures and Tables

**Figure 1 cancers-12-00465-f001:**
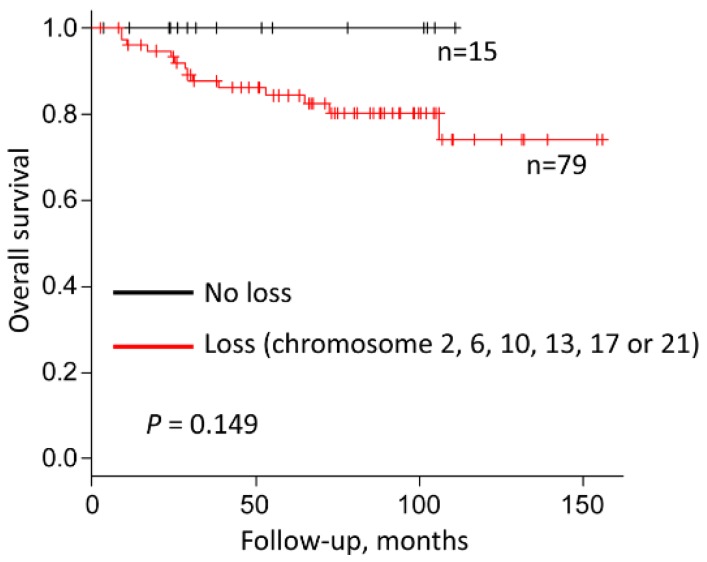
Combined survival analysis of chRCCs categorized by loss or no loss of chromosomes 2, 6, 10, 13, 17 and 21 (TCGA-KICH: No loss *n* = 12; Loss *n* = 52; Swiss cohort: No loss *n* = 3; Loss *n* = 27).

**Figure 2 cancers-12-00465-f002:**
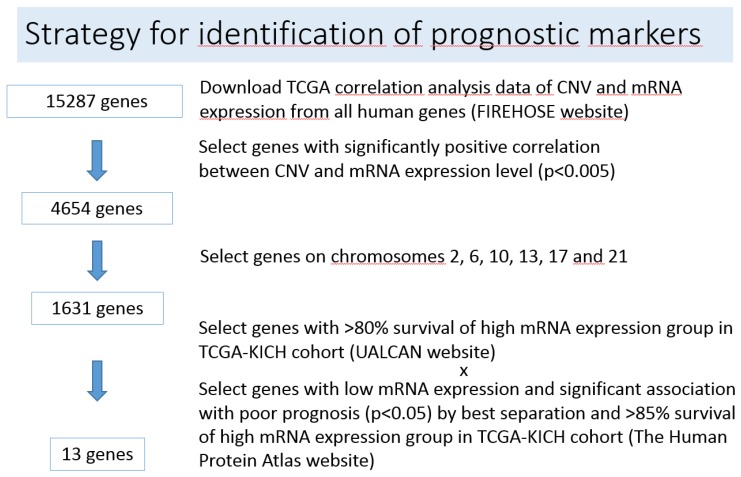
Strategy for identification of prognostic markers.

**Figure 3 cancers-12-00465-f003:**
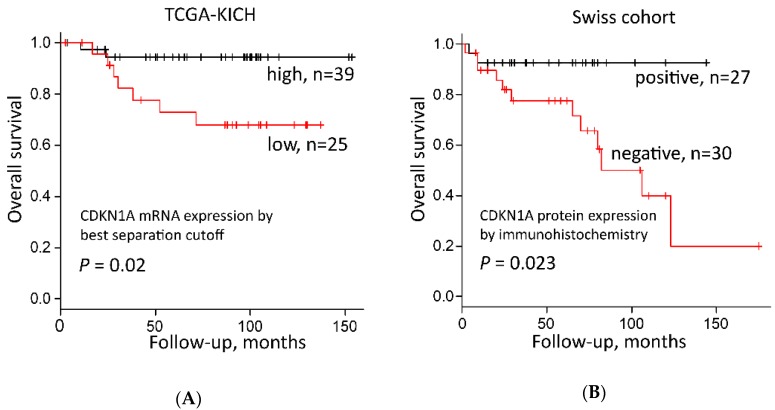
Survival analysis of CDKN1A expression in chRCC. (**A**) CDKN1A mRNA expression and overall survival of 64 chRCC patients in the TCGA-KICH dataset from the Human Protein Atlas [23]—best cut off was according to FPKM values (Fragments per kilo base per million mapped reads); (**B**) CDKN1A protein expression and overall survival of 57 chRCC patients from the Swiss cohort dataset.

**Figure 4 cancers-12-00465-f004:**
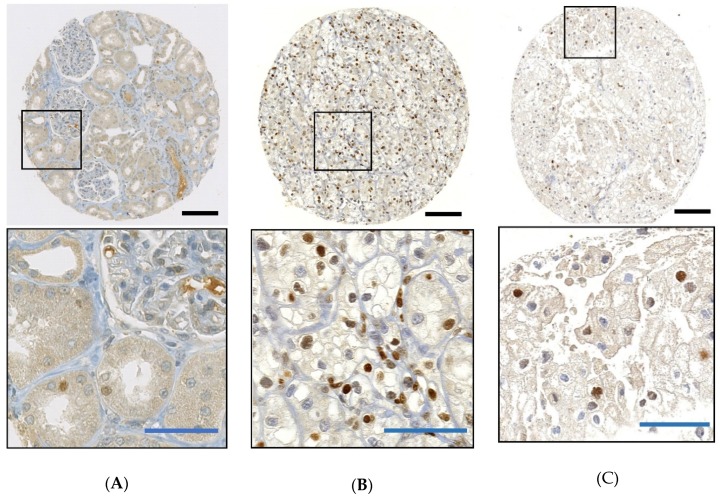
Immunohistochemistry of CDKN1A in the Swiss cohort. (**A**) Weak nuclear CDKN1A expression in some tubular cells in normal kidney; (**B**) strong nuclear CDKN1A expression in clear cell RCC; (**C**) strong nuclear CDKN1A expression in chRCC. Black bars: 100 μm; blue bars: 10 μm.

**Table 1 cancers-12-00465-t001:** Genes with a highly significant correlation between CNV and mRNA expression level, cellular localization of their proteins and their function.

Gene Name	Chromosomal Locus ^1^	CNV vs mRNA Pearson’s Correlation Coefficient ^2^	Protein Expression ^3^	Protein Function (GeneCards ^4^)
*CDKN1A*	6p21.2	R = 0.4434, *p* = 0.0002	nucleus	Cell cycle regulation
*KLF6*	10p15.2	R = 0.5474, *p* < 0.0001	nucleus	Transcriptional activator
*FAM160B1*	10q25.3	R = 0.7632, *p* < 0.0001	cytoplasm	unknown
*PAOX*	10q26.3	R = 0.6088, *p* < 0.0001	cytoplasm	Polyamine oxidase
*PWWP2B*	10q26.3	R = 0.52, *p* < 0.0001	cytoplasm	unknown
*FBXW4*	10q24.32	R = 0.4296, *p* = 0.0003	golgi	Ubiquitination
*FBXL15*	10q24.32	R = 0.4048, *p* = 0.0007	cytoplasm	Ubiquitination
*CASKIN2*	17q25.1	R = 0.4364, *p* = 0.0002	cytoplasm	unknown
*RTN4RL1*	17p13.3	R = 0.4013, *p* = 0.0008	secreted	Brain development
*FMNL1*	17q21.31	R = 0.3974. *p* = 0.001	cytoplasm	Regulation of cell morphology
*RAB37*	17q25.1	R = 0.369, *p* = 0.002	cytoplasm	GTPase
*SOCS3*	17q25.3	R = 0.3611, *p* = 0.003	cytoplasm	Cytokine signaling suppression
*C21orf2*	21q22.3	R = 0.5435, *p* < 0.0001	mitochondria	Regulation of cell morphology, DNA damage repair

^1^ Gene, National Center for Biotechnology Information [26], ^2^ Data from the FIREHOSE, Broad Institute [24], ^3^ Data from The Human Protein Atlas [23], ^4^ GeneCards, The Human Gene Database [27].

**Table 2 cancers-12-00465-t002:** Tumor stage, histological grading according to necrosis and/or sarcomatoid differentiation, CDKN1A expression separated by the best separation cutoff from FIREHOSE [24] mRNA data and overall survival in chromophobe renal cell carcinoma.

Cohort	TCGA-KICH				Swiss Patients			
Variables	Univariate		Multivariate^2^		Univariate		Multivariate	
	HR (95%CI)	*p*-value	HR (95%CI)	*p-*value	HR (95%CI)	*p-*value	HR (95%CI)	*p-*value
Tumor stage (3–4 vs 1–2) ^1^	10.22 (2.12–49.29)	0.004	6.442 (1.488–37.214)	0.012	1.447 (0.398–5.264)	n.s.	1.266 (0.343–4.678)	n.s.
Grade (High vs Low)	18.03 (4.448–73.05)	<0.001	6.087 (1.374–32.266)	0.017	-	-	-	-
CDKN1A expression (Low vs High) ^2, 3^	22.528 (2.862–2904.443)	<0.001	12.527 (1.289–1675.059)	0.026	4.812 (1.07–21.64)	<0.05	4.741 (1.051–21.390)	<0.05

HR, hazard ratio; CI, confidence interval; n.s.: not significant; ^1^ TCGA-KICH: T stage, Swiss patients: pT stage; ^2^ Firth correction was used because of quasi-complete separation; there was no event in one of the subgroups; ^3^
*CDKN1A* mRNA expression in TCGA-KICH cohort and CDKN1A protein expression in Swiss cohort.

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
