# Peer review of "Loss of CDKN1A mRNA and Protein Expression Are Independent Predictors of Poor Outcome in Chromophobe Renal Cell Carcinoma Patients"

_cancers, 2020, doi:10.3390/cancers12020465_

Round 1
Reviewer 1 Report
The work of R. Ohashi et al. is part of a series of studies aimed at discovering new genetic and / or protein markers predictive in cancer and particularly in renal cancer. They clearly demonstrate that loss of CDKN1AmRNA and of the protein expression could serve as a negative prognostic marker in chRC Carcinoma.
Reviewer 2 Report
In this manuscript by Ohashi et al., the authors attempt to identify patterns of gene expression changes associated with survival in chRCC. Among several candidates, they identify CDKN1A as a prognostic marker.
Overall, even though the aim of this study is interesting and relevant to the scientific community, the entire study lacks the accuracy needed to be published. In particular, the study runs shortly in exploring the presented hypothesis, fails to extensively validate the discovered prognostic markers and fails to present strong results supporting the obtained conclusions.
The main critical points I would like to mention are:
In the first part of the study, the authors compare the chromosomal loss and the patient outcome. Interestingly, they find a poor outcome for patients with loss of chromosomes 2,6,10,13,17 and 21. However, the authors do not show the effect of the loss of each of these chromosomes. Since TP53 and PTEN have been previously associated with chRCC, what is the survival of patients with the loss of chromosomes 10 and 17 only (TP53 is on chromosome 17 and PTEN on chromosome 10)? Since CDKN1A is on chromosome 6, what is the effect of the loss of chromosome 6? Overall, a better characterization of the correlation between chromosomal loss and survival rate is needed.
In 2.2., the authors start from a long list of genes present on chromosomes 2,6,10,13,17 and 21 and end up with a list of 13 genes that have low expression and poor prognosis. The strategy used to select genes lacks accuracy. A better description in the method section describing the entire strategy is needed.
Even though the authors present the Pearson’s correlation coefficient, the expression level of these 13 genes is missing.
Can the authors speculate if a connection between these 13 genes is present? Are they part of a common signaling pathway? Are they present in the same patients? Are they mutually exclusive?
In 2.3., among the 13 genes, the authors focus on CDKN1A. Why did the authors focus on CDKN1A? Genes such as KLF6 have a stronger CNV vs mRNA correlation coefficient according to Table 1. Thus, according to the data presented by the authors in Table 1, KLF6 is a stronger candidate for validation. More importantly, well known is the relation between TP53 activation and CDKN1A in tumors. However, the authors did not mention whether the levels of expression of CDKN1A in chRCC are correlated to the level/mutation status of TP53. This type of correlation will be important to better understand Figure 3 and Table 2. Additionally, immunohistochemistry data is presented in Figure 4C. However, a single representative image of CDKN1A positive chRCC is shown. This data lacks the accuracy required for publication. At least quantification of each performed immunohistochemistry should be presented in a graph.
Reviewer 3 Report
General comments
This paper discusses about predictors of chromophobe renal cell carcinoma (ChRCC). The main contribution of the paper is to demonstrate low expression of CDKN1A on chromosome 6 as well as deletion of chromosomes No 2, 6, 10, 13, 17 and 21 worsened overall survival rate. This study is well-designed, and the methods are reasonable and sophisticated. I think that the paper has some values for publication in the journal after some revisions since the histological predictors of ChRCC remain to be completely elucidated.
Major points
The authors should indicate the reasons why you analyzed only CDKN1A without analyses for the other 12 genes. Additionally, you should discuss about the contributions of the 12 genes for overall survival rate.
Minor points
1. The authors should show reason why they set 2% as cut-off value of CDKN1A immunohistochemical assessment.
2. The authors only indicate stage and CDKN1A as variables in Table 2. Did you only use two variables in multivariate analysis? You need to describe all variables if you analyzed other variables.
3. For case selection from TCGA-KICH cohort, the authors should show cut-off values for assessment of mRNA expression and CNV. How do you classify mRNA low expression or mRNA high expression. Additionally, you should present what events were regarded as "poor prognosis".
Reviewer 4 Report
The Authors reported loss of CDKN1A mRNA and protein expression as an indipendent predictors of poor outcome in chromophobe renal cell carcinoma patients.
Minor:
-Immunohistochemical expression: are there potential heterogeneity on relevance of p21 expression on whole tumour slides vs punched TMA biopsies?
Please insert a comment.
-are there observable data related to progression-free survival in addition to overall survival? If yes, please insert observations.
-p21 clones: insert a comment which higlight differences on clones related to p21 immunoreactions. Historical discordant data may be associated to clones rather than clinical different tumour setting.
-chromophobe renal cell carcinoma may show the tumor dormancy phenomena (metastases after many years after primary diagnosis) usually characterized by single metastases or multiple metastases and/or sarcomatodi differentiation. Please specifiy if possibile the cohort of the study (Swiss).
Round 2
Reviewer 2 Report
Overall, after the revision, the study has much more accuracy in explaining and exploring the presented hypothesis. It is thus improved in explaining the reader the importance of the study and the importance of investigating the role of CDKN1A in RCC.
However, I strongly suggest the authors to include more than a representative image of the 124 CDKN1A positive chRCC shown in Figure 4. I understand that a quantification may be complicated, but it is important to provide strong evidences supporting this major conclusion. Presenting a representative image is scientifically unacceptable and thus a strong limit to strongly support the acceptance of this work.
Moreover, although I understand that the other markers besides CDKN1A may be complicated to validate, I would like to see a closing statement to suggest how the other markers may be explored in the future as possible molecular marker to detect chRCC patients.
Author Response
As suggested by the reviewer, we quantified the IHC expression of CDKN1A with H-score with graphs as shown in supplementary Figure S5 and modified 2.3 Result section from line 126 to 138, and 4.4 Materials and Methods section accordingly. Moreover, we added a closing statement on biomarker testing to detect aggressive chRCCs.